# Presenting Health Status in Children Using a Radar Plot

**DOI:** 10.3390/sports8040053

**Published:** 2020-04-23

**Authors:** Asgeir Mamen, Lars Erik Braaum, Per Morten Fredriksen

**Affiliations:** School of Health Sciences, Kristiania University College, Prinsensgate 7-9, 0152 Oslo, Norway; larserik.braaum@kristiania.no (L.E.B.); permorten.fredriksen@kristiania.no (P.M.F.)

**Keywords:** health, physical activity, exercise

## Abstract

Background: To try out the feasibility of presenting the health status of children 6 to 12 years old by using radar plots. Methods: With data from the Health Oriented Pedagogical Project (HOPP) we have described the health status for 1340 children aged 6 to 12 years. We collected or calculated: stature, body mass, waist circumference, waist to height ratio, high density lipoprotein (HDL) and total cholesterol concentration, blood pressure, accelerometer assessed physical activity, endurance interval running performance, and quality of life. Pertinent variables were presented through a radar plot for both individual cases and groups. Results: The boys showed better endurance and recorded more moderate to vigorous physical activity than the girls. The activity level dropped from age 6 to age 12 for both sexes. The girls showed a lower systolic blood pressure compared with boys. Self-rated quality of life was high among boys and girls. Conclusions: This cohort showed good health and the radar plot made it easy to visualise health status for groups and individuals.

## 1. Introduction

The global burden of disease involves a variety of factors with the ability to increase the risk of illness. Occurring alone, each factor poses a potential threat to health, while in combination, the magnitude of risk increases significantly for chronic illness and early fatal death. It is well documented that such risk factors are established in childhood, partly due to genetic factors but are also due to lifestyle-related behaviour. In the case of children, this is mainly associated with parental choice of lifestyle. As an example, it has been discovered that atherosclerosis can be established in young children, as found in the Bogalusa Heart Study [1].

Giving an accurate description of health status can be difficult. During the Crimean war in the 1850s, Florence Nightingale pioneered this presentation by using coxcomb graphs to illustrate the grim health situation in military hospitals [2]. Today there are several graphs available for the researcher. A radar plot, invented by Georg von Mayr, 1877 [3] can be viewed as a connected line graph, thereby reducing the size of the plot. The circular form makes it easy to compare different entities, especially if there is an agreed sequence of the variable, and, as we have done, included a reference line. It has been found useful in social sciences and health sciences to illustrate development or differences. Noirhomme-Fraiture suggests using such a plot for large data analysis [4]. The plot is also known as a Spider Plot or Star Plot among other names.

Although to date, radar plots have not been widely employed in health sciences, they demonstrate potential as an effective means to visualise multiple health related variables in one plot [5]. This is useful as noted in recent conference proceedings [6] where Ordóñez et al. discuss several plots, among them 2D and 3D radar plots for presentation of data in an Intensive Care Unit. Another example is a study using radar plots to describe changes in European Union (EU) alcohol policy [7], showing the broad spectre of data such a plot can handle. The present study demonstrates the potential usefulness of radar plots in visualising health related variables, with data representing a large number of participants and comprising carefully chosen and clinically relevant variables. Despite this, it should be noted that the sample in the present study does include use of some self-report data from children which should be considered a limitation.

However, waist to height ratio (WHtR) does not take into account other variables of importance for good health, such as blood pressure, blood lipids, and social variables such as quality of life. There would therefore seem to be a need for a more detailed description of children’s health status; however, one might ask why healthy children need a health profile. In view of the growing number of children with lifestyle-related risk factors like overweight, inactivity, and low quality of life, an easy to use and comprehensible health profile instrument may be of help when mapping both status quo and intervention effects. A health profile that provides full details of the physical and mental health of an individual and a population would be of immense help for both the public health care system and the individual general practitioner. If the health profile also makes it easy to perceive changes in an intuitive way, we believe most health care personnel would be willing to use such a profile.

A comprehensive overview of an individual’s physical and mental state may be one tool to counteract some of the health problems facing a large part of the world’s population. To create such an overview, several variables may be perceived as important for constructing a health profile. Physical variables, such as anthropometric data, physical performance measures and physical activity level, and medical variables, such as blood pressure and results of various blood tests, are all important. In addition, softer data such as quality of life, emotions, self-esteem, friendship, nutrition, and ability to concentrate could also form part of such a profile.

To prevent lifestyle-related diseases in childhood and adolescence, an individual health profile is crucial for revealing changes. Such a profile should be related to established reference values for the age group and gender and possibly be standardised with respect to variables and units.

The aim of this study is to present a suggestion for a health profile for children that we believe is easy to interpret and that provides a more comprehensive synopsis of essential factors for health than a single measurement.

## 2. Materials and Methods

### 2.1. Sample

The study population was recruited from the Health Oriented Pedagogical Project (HOPP), a controlled longitudinal school-based physical activity intervention programme. The project is registered as a clinical trial (ClinicalTrials.gov Identifier: NCT02495714) and accepted by the Regional Ethical Committee (REC Identifier 2014/2064) and is described in detail elsewhere [8] Shortly, the children of the elementary schools in Horten Municipality (n = 7) were given 45 min of extra physical activity in connection with teaching activities each day. The evaluation of the effect of this intervention continues until 2021 with yearly collection of health, academic, and sociological data from the participating children. As control schools, two schools in the Greater-Oslo area participated, N = 2297. Informed consent from parents/guardians was obtained before the data collection started. The children can leave this evaluation project at any time, but not the school programme as that is the mandatory school programme for the public schools in the municipality. Parents and children aged 6–12 years from nine schools received an invitation to participate in the project. Of a population of 2816 children, 2297 children (82%) were allowed to participate by their parents. The children were given the opportunity to opt out of the blood sample test, while participating in the other test procedures, and 58.4% of the children (n = 1340, n = 650 girls) provided blood samples. Age was defined as age on the test date. The ethnicity of the children was not recorded; however, most of the children were of Caucasian origin. The variables were collected in spring 2015, which are the baseline results.

### 2.2. The Radar Plot

As a radar plot can handle several variables in a comparable state, such a plot was chosen to present the health profile. To facilitate comparison, variables were presented as percentages of the reference values. To provide a uniform view, the scales were arranged so that all values better than the reference are higher values. The lower the value, the more negative the result. This will aid the viewer in seeing that variables closest to the centre are those that need most attention, to enable groups or individuals to be compared to the reference value. All graphs have an age/gender-dependent median value for all variables.

Table 1 lists the chosen variables with reference values and how they are presented.

### 2.3. Measurements

#### 2.3.1. Blood Pressure

Blood pressure was measured with an Omron M6 IT (Omron Healthcare Co, Ltd, Kyoto, Japan) with the child seated and with the left arm resting on a table. In Norway for children aged 7–12 years, the mean systolic blood pressure was found to be between 95 and 115 mmHg, and we use this as a reference for our data [9]. A risk limit may be hard to define in children as blood pressure increases naturally with increasing body mass, increased hydrostatic pressure, and hormone changes. However, we used the highest 90th percentile as cut-off point for each age group and gender. Any value below that would be considered not harmful. For the visual effect, the scale was inverted, so a low value has a high score.

#### 2.3.2. Waist Circumference

Waist circumference (WC) was measured according to the WHO STEPS standard where measurements are made at the approximate midpoint between the lower margin of the last palpable rib and the top of the iliac crest after a normal expiration [10]. WC was found to be a better predictor of obesity-related health risk than body mass index (BMI) [11] It is also well established that adiposity around the waist is considered to be more harmful with regard to cardio-metabolic diseases than fat distributed around the body [12] The International Diabetes Federation has suggested using the 90th percentile for children aged 10 to 16 [13] and we have followed this recommendation in our plot.

#### 2.3.3. Waist to Height Ratio (WHtR)

The WHtR was calculated as the WC in cm divided by the height in cm. Height was measured with a portable Seca 213 stadiometer (Seca GmbH, Hamburg, Germany). The subject stood with feet together without shoes, looking straight ahead. Height was recorded to the nearest 0.5 cm. WHtR was found to correlate better with abdominal obesity and cardio-metabolic risk factors than traditional BMI and WC in children [14]. WHtR is considered to be more useful than BMI as it is better at locating adiposity, does not involve other tissue, and may be used across age and gender [15]. Against this background, WHtR was chosen to characterise overweight problems in the population, with a cut-off ratio of 0.5 as suggested by [16]. This ratio was proven to correspond relatively well with BMI cut-off ratios with regard to the risk of cardio-metabolic diseases [17]. The scale was inverted to achieve a consistent visual effect.

#### 2.3.4. Quality of Life (QoL)

In any health profile, a variable describing an individual’s well-being should be included. Self-reported quality of life (QoL) was assessed with the Norwegian version of the Inventory of Life Quality in Children and Adolescents, (ILC) [18]. This is a seven-item inventory that uses 5-point Likert scales to score the answers. Children below 11 years were advised to do the test by interview. We did this by having the children fill out the questionnaire themselves, under supervision. The inventory was given both to children and to parents. The parental score is not presented here. Highest achievable score, 100, was used by us as the reference score. Kristensen and Hove [19] evaluated the internal consistency of the Norwegian version of ILC and found it to be acceptable (Cronbach α 0.63. to 0.76). Reliability was found to be good, 0.72).

#### 2.3.5. Blood Samples

Blood was collected in a nonfasting state during the school day by trained phlebotomists. Blood was drawn from the antecubital vein in 4 mL serum tubes with a clot activator (Vacuette®, Greiner Bio-One, Kremsmünster, Austria) and analysed in the central laboratory of Vestfold Hospital Trust. Total cholesterol and HDL cholesterol were analysed on Vitros 5.1 (Ortho-Clinical Diagnostics, Franklin, Raritan, NJ, USA) with reagents from the supplier.

#### 2.3.6. Serum Cholesterol

The American Academy of Pediatrics defines serum cholesterol above 5.2 mmol as pathological for children aged 6–12 [20]. Here, values lower than 5.2 are displayed as higher values on the radar diagram.

#### 2.3.7. High Density Lipoprotein (HDL)

High levels of HDL were shown to correspond with reduced risk of cardiovascular diseases in adults. By extrapolation, it is thought that high values are preferable also in children. The American Academy of Pediatrics suggests a value above 1.66 mmol/L to be “acceptable” and below 1.04 to be “low” [21], but there is no national or international consensus regarding a preferred and clinically positive cut-off value for HDL. The reference value in this diagram is 1.5 mmol/L. Hence, higher values are positive.

#### 2.3.8. Physical Activity (PA)

The level of physical activity was monitored by Actigraph wGT3X-BT accelerometers (ActiGraph LLC, Pensacola, FL, USA). Participants were instructed to attach the device to the right side of their hip with an elastic band for seven consecutive days, at all hours, unless injured, ill, showering, swimming, or absent from school on the day of testing. A detailed description of the accelerometer recordings is given in [22]. PA levels were based on mean counts per minute (cpm) with 0–99 cpm as sedentary, 100–1999 cpm light, 2000–4999 cpm moderate, and ≥ 5000 cpm vigorous. The traditional minimal standard given for children is 60 min of PA a day, but it was suggested that 90 min a day with a moderate to vigorous intensity level would be a better goal [23]. The reference value was therefore set to 90 min a day.

#### 2.3.9. Endurance Test

Several studies have shown that enhanced aerobic fitness may be a significant contributor in reducing the risk of acquiring cardio-metabolic diseases in both children [24] and adults [23]. Several shuttle run tests exist, but the Andersen interval shuttle run test [25] was especially developed and designed for children. In the Andersen test, the participants perform a shuttle run between two cones 20 m apart at self-selected pace for 15 s, then rest for 15 s and continue the test for 10 min. The distance covered is used here as the performance criterion. The test was found valid and with an acceptable reliability for 10-year-old children [26]. The 90th percentile for each age group was used as reference value. Hence the higher the value, the better protected the child was against cardio-metabolic diseases.

### 2.4. Statistics

Results are presented as mean (SD) unless otherwise stated. The radar graphs present the median of each age-gender group. As reference, either age-gender specific values, generally accepted cut-off values, or normative data were used; see Table 1 for details. Independent groups were compared with Gosset (Student) t-test. The statistical software used included IBM SPSS v. 25 (IBM, Armonk, NJ, USA) and SigmaPlot v. 14 (Systat Software, GmbH., Erkrath, Germany).

## 3. Results

### Basic Results

Height and body mass increased normally with age, as 6th graders were 29 cm taller than 1st graders and weighed 18.7 kg more. WHtR did not change and was between 0.44 and 0.47 across the age span. WC increased by 9.9 cm across age for girls and by 11.9 cm for boys. The increase in systolic blood pressure was slightly smaller among girls, but values were within the normal range.

The concentration of HDL was significantly higher in boys across age (p < 0.05), compared with girls. On the other hand, total cholesterol values did not differ much between the genders or age groups.

Physical activity, measured as moderate to vigorous physical activity (MVPA), showed typical gender and age development, with boys having higher scores and with a decrease in activity level with increasing age for both genders. MVPA decreased by 18 min a day from 1st to 6th grade for boys and by 20 min for girls. Endurance performance, measured using the Andersen interval shuttle run test, showed that boys ran farther than girls did in all age groups. The biggest average difference was in 6th grade, at 41 m. The increase in maximal running distance developed rather linearly with age, but between 3rd and 4th grade, the development plateaued for girls. For boys, running distance increased by 49 ± 27 m·year^−1^ and for girls by 45 ± 25 m·year^−1^. With regard to quality of life, the children seemed quite satisfied as both genders revealed a small, but statistically significant, increase in well-being across grades: see Table 2.

In Figure 1, the children scored below reference values on systolic blood pressure and the Andersen Test. This is because the reference values are the 90th percentile, and the scores are the median value. For the variables that use a fixed reference value: HDL, total cholesterol, MVPA, and quality of life, the scores were often better than the reference. The radar graph makes it easy to pinpoint specific variables and how the scoring of these relates to the reference and also to detect gender differences.

Figure 2 shows two cases of children considered to be at risk for future cardiovascular disease. They both score well below their classmates’ values and the reference line on several of the variables. The plot provides an easily interpretable view of the reduced health score of these individuals. 

## 4. Discussion

### 4.1. Main Findings

The children from the HOPP study appear to be of good, normal health both physiological and mental. The most worrying factor may be the significant decline in physical activity level as age increases. On the positive side were the blood tests, where the values were consistently better than the Norwegian reference values. A radar graph made it easy to visualise the overall health status for both groups and individual cases.

### 4.2. Choice of Variables

The HOPP project produces a plethora of variables and selecting significant variables to represent a suitable health profile proved difficult. Not all variables were measured in all age categories in the HOPP project. Our choice of variables was therefore based on measurements performed on all age groups and based on the theoretical and empirical explanatory strength of each variable on health. Most variables concerning children’s lifestyle-related risk factors are based on an extrapolation from adulthood. The variables should thus only be seen as risk factors for future probability of illness or disease and not as definitive endpoints with regard to children, as not all factors are traceable into adulthood. There is an ongoing concern regarding disagreement about risk factor determinants. This hampers comparison across scientific findings, thus making it difficult to draw conclusions about the epidemiological situation [27]. Paediatric cut-off values are generally problematic and questionable as their future health consequences are poorly validated. Definitions of paediatric risk factors should therefore be applied with caution. However, some longitudinal studies have revealed that classification of childhood risk factors does predict adult metabolic syndrome and type 2 diabetes [28]. 

### 4.3. Anthropometric Variables

Table 1 displays the chosen anthropological variables from grade 1 to 6 for both sexes. The development was normal for height and body mass, as we presented earlier [29]. WHtR did not change over time in either boys or girls. No clear trend was seen, and the largest change was only 0.03, for girls in first grade (WHtR = 0.47) to sixth grade (WHtR = 0.44). This would indicate that obesity problems were not a serious issue in the cohort, as Zhou et al. found WHtR to be a simple and practical screening tool for obesity and metabolic syndrome in children [30]. Letho et al. found both WC and WHtR to be associated with children’s health behaviour [31]. We found a significant correlation between average MVPA and WHtR (*p* = 0.049, not shown), but the significance is due to a large N. On a grade/gender level, boys in 5th grade had the highest correlation, with r = 0.22, which corresponds to a small effect size. One should also be critical of the accuracy of WHtR to detect health status. Sardinha et al. found low precision for WHtR and BMI to classify increased CVD risk [32]. Further, Keefer et al. could not find an association between systolic blood pressure, total cholesterol, and WHtR for the young segment of the National Health and Nutrition Examination Survey (NHANES) cohort but found a significant association in the entire cohort [33]. This weakness of single variables to predict CVD health is an important argument for using a carefully selected blend of variables for risk determination.

### 4.4. Medical Variables

Blood pressure is by nature difficult to measure due to its large variability caused by movement during measuring, disturbances catching the children’s attention, emotional variability of any kind, and PA of high intensity prior to measurement. The latter was avoided by placing restrictions on PA two hours prior to blood pressure testing. Usually blood pressure is measured three times and the mean value is used. In the present study, this was not possible; however, the mean systolic blood pressure values are still within the normal range. The systolic blood pressure (SBP) was within the normal range for Norwegian children for both sexes and all age groups. The increase seen from first to sixth grade does not deviate much from reference data of Norwegian children [34]. All children with initial high systolic blood pressure values were retested about four weeks later, and no child was found with pathological values requiring treatment. The reason for using systolic and not diastolic is the large variability and uncertainty in the measurement of diastolic blood pressure. It is unusual for children to have high blood pressure, which might suggest that a persistent high reading in a child is significant and a clear indication for treatment with nutrition changes and increased PA, or in some cases, antihypertension medication.

#### 4.4.1. HDL

The HDL concentration did not differ across age but it did between genders. One reason might be that the higher PA level in boys has a positive influence on HDL concentration, as was found in other studies [35]. If so, the reduced levels of PA observed with age are a cause for concern.

#### 4.4.2. Total Cholesterol 

Studies have established atherosclerosis in children as young as two years old, and an elevated level of serum cholesterol may increase the risk of further development of the disease [36]. The present results show normal values, however, Strand et al. showed in a previous paper from the HOPP study that some children do display high values [37]. Approximately 7% of the children in the present study had values above 5.2 mmol/L, which is considered pathological [38]. No statistical differences were found between total cholesterol levels and sex, except for girls in 2nd and 4th grade who had 4.27 (0.66) vs. 4.55 (0.69) mmol/L respectively (p = 0.04). How much this difference matters for future cardiovascular health is uncertain. The effect size is 0.41, indicating low clinical value.

### 4.5. Activity Level

#### MVPA

A high physical activity level as a child may track positively into adulthood [39]. It is therefore important to maintain a high activity level in children throughout childhood, by engaging them in motivating activities, such as play and games. The reduction in activity level was substantial for both boys and girls, with 18 and 20 min reduced time in MVPA from 1st to 6th grade, which corresponds to a 20% decrease in activity. Only 1st–4th grade boys and 1st–3rd year girls had MVPA above the recommended 90 min a day [24]. The reduction in activity seen when using accelerometers might also include measurement bias, as height increases with age, increasing the time of each leg movement, and thus less movement is recorded. In addition, from an energy expenditure point of view, it may be difficult to compare young and older children, as their increase in body mass will make it more demanding to move a larger body [40]. In addition, longer legs are like a longer pendulum, and thus reduce acceleration and give lower values [21]. Hence, even if accelerometer readings show a decline, this may not affect energy expenditure to the same degree.

### 4.6. Endurance Performance

As expected, running performance on the Andersen Test increased with age. Compared with the values for 10-year-old children in the study by Aadland et al. [26], our results are significantly better for the 5th graders. For boys this difference was 35 (16.5) m, p = 0.04, while for girls it was 66 (14.5) m, *p* < 0.01. It may be difficult to make valid comparisons between measurements taken at different locations. The results of the HOPP study are a blend of results from nine different schools, which all have different surfaces to run on. Further, the HOPP study used different data collectors on the schools and the children had no identification number on them when running. Misclassifications may thus be a source of error and may partly explain the observed difference.

### 4.7. Quality of Life 

Quality of life is considered an important variable for evaluating the overall health status of a child. Low scores on quality of life may represent a significant threat to child and adolescent health and may cause physiological problems and vice versa. Scores from the present study on life quality were high, indicating that these children have a good life. The score steadily increased a little year by year and the scores in first grade were statistically lower than the 4th to 6th grade scores for both boys and girls.

Despite being overweight and in poor health, the girl used as an example in Figure 2 scored high on quality of life, better than the median of her group. Even though many children experience negative feedback and cultural stigma due to overweight and poor physical health, life quality for this girl is evidently more than just physical health. This is a good illustration of the need for “softer data” in a health profile.

### 4.8. Strengths and Limitations

The strengths of this study are the large number of participants and the high number of carefully chosen variables collected. The weaknesses are the large number of persons that took part in the data collection and the use of some self-report data from children. 

## 5. Conclusions

The investigated population was generally in good CVD health. A radar plot was feasible to display a large number of health-related variables in a child population. The plot was useful in displaying both group data and cases. The presentation with a radar plot can help to identify children, or groups, in need of early intervention and may be used to investigate health tracking through childhood to adolescents and adulthood. If such a presentation is found useful, an expert panel should decide which variables and reference values should be used, any weighting of variables, and how the data should be presented.

## Figures and Tables

**Figure 1 sports-08-00053-f001:**
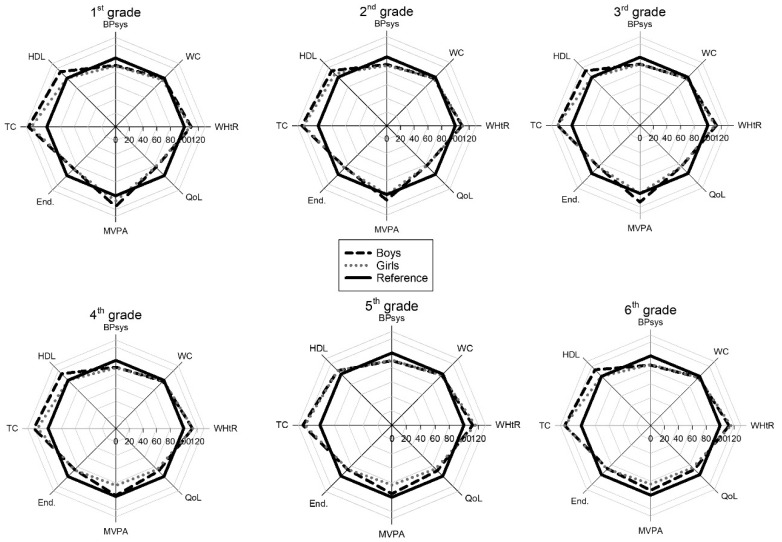
Health profiles for boys and girls from 1st to 6th grade. The higher the score, the healthier the value. WHtR = Waist to Height Ratio, MVPA = Moderate to Vigorous Physical Activity, Tot Chol = Total Cholesterol, HDL = High Density Lipoproteins.

**Figure 2 sports-08-00053-f002:**
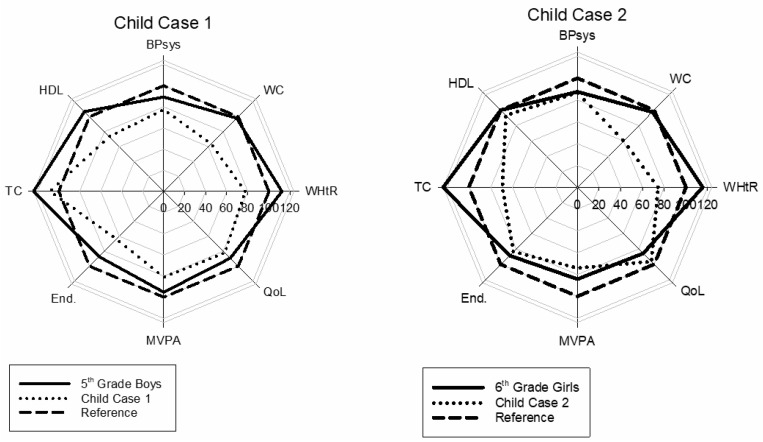
Two child cases to illustrate the health profile. Legends as in Figure 1.

**Table 1 sports-08-00053-t001:** Variables and reference values used in the radar plot.

Variable	Unit	Reference Value	Presentation
Systolic Blood Pressure	mmHg	90th percentile	Inverted †
Waist Circumference	cm	90th percentile	Inverted
Waist to Height Ratio	-	0.50	Inverted
Quality of Life	%	100	Normal
Moderate to Vigorous Physical Activity	Average/min/day	90	Normal
Endurance Test	m	90th percentile	Normal
Total Cholesterol	mmol/L	5.2	Inverted
High Density Lipoprotein Concentration	mmol/L	1.5	Normal

† Inverted means that a low raw score is presented with a high value, and a high raw score with a low value. Since low blood pressure is healthier, the scale is inverted. A high value on the endurance test is preferable, so a high score gives a high value and the scale is normal.

**Table 2 sports-08-00053-t002:** Values for each age group sorted by gender.

Girls							
Grade		1	2	3	4	5	6
Height (m)	mean	1.24	1.29	1.34	1.40	1.47	1.53
	SD	0.06	0.06	0.06	0.07	0.08	0.08
	10th percentile	1.16	1.22	1.27	1.31	1.37	1.42
	90th percentile	1.30	1.37	1.42	1.48	1.57	1.62
Body Mass (kg)	mean	24.3	26.8	30.2	33.7	38.3	43.0
	SD	4.8	5.3	6.0	8.0	8.7	9.8
	10th percentile	19.2	21.5	23.1	25.8	29.3	32.3
	90th percentile	30.0	33.5	37.1	44.0	50.2	56.2
WHtR	mean	0.47	0.46	0.46	0.45	0.45	0.44
	SD	0.04	0.04	0.04	0.05	0.05	0.06
	90th percentile	0.42	0.42	0.41	0.39	0.40	0.39
	10th percentile	0.53	0.52	0.51	0.50	0.51	0.52
WC (cm)	mean	57.8	59.3	61.5	63.0	65.5	67.7
	SD	6.2	5.9	6.6	8.6	8.1	9.0
	90th percentile	51.1	54.0	54.0	54.0	57.6	59.0
	10th percentile	66.0	67.1	70.0	72.7	75.5	79.0
SBP (mmHg)	mean	104	106	107	108	111	112
	SD	10	11	10	11	10	11
	90th percentile	92	92	96	96	100	98.3
	10th percentile	117	119	121	122	123	125.0
HDL (mmol/L)	mean	1.54	1.63	1.56	1.56	1.59	1.57
	SD	0.33	0.36	0.33	0.33	0.34	0.32
	10th percentile	1.10	1.1	1.20	1.20	1.16	1.20
	90th percentile	2.03	2.1	2.00	2.00	2.00	2.00
Tot Chol (mmol/L)	mean	4.31	4.27	4.33	4.55	4.38	4.26
	SD	0.64	0.66	0.71	0.69	0.69	0.64
	90th percentile	3.60	3.50	3.50	3.60	3.50	3.50
	10th percentile	5.10	5.20	5.19	5.53	5.30	5.20
Andersen Test (m)	mean	779	855	934	934	975	1022
	SD	121	104	127	116	101	112
	10th percentile	610	715	790	797	843	890
	90th percentile	949	982	1085	1078	1125	1155
MVPA (min)	mean	98	91	91	79	79	78
	SD	26	26	26	25	28	24
	10th percentile	64	57	60	50.2	46	47
	90th percentile	132	126	128	113	120	113
QoL (%)	mean	80.8	81.7	83.3	84.2	85.4	85.3
	SD	11.8	10.2	11.5	9.2	9.5	10.7
	10th percentile	64.3	67.9	67.9	71.4	71.4	71.4
	90th percentile	96.4	96.4	96.4	96.4	96.4	96.4
**Boys**							
**Grade**		**1**	**2**	**3**	**4**	**5**	**6**
Height (m)	mean	1.24	1.30	1.36	1.41	1.47	1.53
	SD	0.06	0.06	0.06	0.06	0.08	0.08
	10th percentile	1.16	1.23	1.29	1.33	1.37	1.42
	90th percentile	1.31	1.38	1.44	1.49	1.57	1.62
Body Mass (kg)	mean	23.8	27.3	31.1	35.0	39.0	42.5
	SD	4.3	4.1	5.8	6.9	8.0	8.5
	10th percentile	19.4	22.7	25.0	27.3	30.9	33.3
	90th percentile	29.4	33.1	37.6	45.2	49.8	54.2
WHtR	mean	0.46	0.46	0.45	0.46	0.46	0.45
	SD	0.03	0.04	0.04	0.04	0.06	0.05
	90th percentile	0.42	0.43	0.41	0.39	0.41	0.40
	10th percentile	0.50	0.51	0.52	0.50	0.54	0.51
WC (cm)	mean	56.9	59.9	61.5	64.5	67.1	68.8
	SD	5.0	5.3	6.6	7.4	8.4	8.0
	90th percentile	51.1	54.5	54.0	56.5	59.0	60.0
	10th percentile	63.0	67.0	70.0	75.0	79.0	80.0
SBP (mmHg)	mean	104	107	110	109	112	113
	SD	9	10	9	10	9	11
	90th percentile	93	95	98	98	100	98
	10th percentile	116	118	122	120	123	127
HDL (mmol/L)	mean	1.67	1.71	1.71	1.70	1.62	1.66
	SD	0.39	0.35	0.37	0.33	0.36	0.35
	10th percentile	1.25	1.24	1.20	1.30	1.20	1.20
	90th percentile	2.25	2.10	2.30	2.10	2.10	2.10
Tot Chol (mmol/L)	mean	4.31	4.24	4.33	4.30	4.30	4.24
	SD	0.66	0.59	0.71	0.65	0.586	0.634
	90th percentile	3.40	3.60	3.50	3.46	3.60	3.50
	10th percentile	4.90	5.06	5.19	5.04	5.20	5.00
Andersen Test (m)	mean	806	870	966	995	1009	1063
	SD	112	118	126	132	120	111
	10th percentile	641	740	790	849	840	915
	90th percentile	949	1014	1116	1160	1160	1215
MVPA (min)	mean	106	98	102	92	89	88
	SD	26	26	28	29	30	28
	10th percentile	73	65	67	76	54	57
	90th percentile	136	135	139	135	130	128
QoL (%)	mean	81.0	81.1	83.7	86.0	86.0	86.0
	SD	13.08	13.0	9.9	9.3	9.3	9.3
	10th percentile	62.93	64.3	70.4	75.0	75.0	75.0
	90th percentile	97.9	96.4	96.4	96.4	96.4	96.4

WHtR = Waist-to-Height Ratio, WC = Waist Circumference, SBP = Systolic Blood Pressure, Tot Chol = Total Cholesterol, MVPA = Moderate to Vigorous Physical Activity, QoL = Quality of Life.

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
