# Peer review of "Presenting Health Status in Children Using a Radar Plot"

_sports, 2020, doi:10.3390/sports8040053_

Round 1
Reviewer 1 Report
This study tried out the feasibility of presenting the health status of children 6 to 12 years old by using Radar Plots. As a result of the review, I have decided to major revision on the publication of the manuscript in MDPI.
Major concern
Introduction section (lines 24–67)
- This study used Radar Plot to present health status in children, but there is no any description of Radar Plot in introduction section. Why radar plot of many ways should be used? Are there any other advanced studies that used Radar Plot in similar research design? Rationale for using Radar Plot should be described in detail.
Subjects section (lines 70–73)
- Even though reference is suggested, more information about HOPP should be described. In addition, it is doubtful how the boys understood and agreed to participate in the research.
Blood pressure section (lines 99–101)
- The subject of this study was 6-12 years old according to sample section. “For children aged 7-12 years, mean systolic blood pressure is between 95-115 mmHG”. I am confused whether the age was 6-12 or 7-12, if there is a reason for changed age, please describe it.
Results section (lines 183–201)
- It is said that height and body mass increased normally with age, as 6th graders were 29 cm taller than 1st graders and weighed 18.7kg more. Simply suggesting descriptive statistics, but I recommend other statistical method, such as sampling significant variables and comparing it using t-test to find something significant result.
Discussion section (lines 224–341)
- The most significant differences suggested in the Results section were re-described, and no in-depth discussion such as a comparison with advanced studies or mechanisms. It simply suggested the utility of radar graph, but there should be comparison of advanced studies, or other plot and graph. Also, the relationship between those plot or graph and measured variables should be described.
Minor concerns
(1) lines 231 and 245, (2) lines 263 and 275, (3) lines 293–305, (4) lines 307–315, and (5) line 317-329
- Comparison with advanced studies should be added in various aspects, especially with regard to mechanism and adult health status.
Author Response
We thank You for comments! A respons to each is presented here:
Major concern
Introduction section (lines 24–67)
- This study used Radar Plot to present health status in children, but there is no any description of Radar Plot in introduction section. Why radar plot of many ways should be used? Are there any other advanced studies that used Radar Plot in similar research design? Rationale for using Radar Plot should be described in detail.
We acknowledge this remark and have now included a paragraph describing the history of the plot and how/when it can be useful.
Giving an accurate description of health status can be difficult. During the Crimean war in the 1850’s, Florence Nightingale pioneered this presentation by using coxcomb graphs to illustrate the grim health situation in military hospitals[2]. Today there are several graphs available for the researcher. A Radar Plot, invented by Georg von Mayr, 1877[3] can be viewed as a connected line graph, thereby reducing the size of the plot. The circular form makes it easy to compare different entities, especially if there is an agreed sequence of the variable, and, as we have done, included a reference line. It has been found useful in social sciences and health sciences to illustrate development or differences. The plot is also known as a Spider Plot or Star Plot and other names [4].
Subjects section (lines 70–73)
- Even though reference is suggested, more information about HOPP should be described. In addition, it is doubtful how the boys understood and agreed to participate in the research.
We acknowledge this comment and have made an amendment to the paragraph in question:
Shortly, the children of the elementary schools in Horten Municipality (n=7) are given 45 min of extra physical activity in connection with teaching activities each day. The evaluation of the effect of this intervention continues until 2021 with yearly collection of health, academic, and sociological data from the participating children. As control schools, two schools in the Greater-Oslo area participate. Informed consent from parents/guardians were obtained before the data collection started. The children can leave this evaluation project at any time, but not the school programme as that is the mandatory school programme for the public schools in the municipality.
Blood pressure section (lines 99–101)
- The subject of this study was 6-12 years old according to sample section. “For children aged 7-12 years, mean systolic blood pressure is between 95-115 mmHG”. I am confused whether the age was 6-12 or 7-12, if there is a reason for changed age, please describe it.
We acknowledge this point. We have tried to re-phrace the scentence:
Blood pressure was measured with an Omron M6 IT (Omron Healthcare Co, Ltd, Kyoto, Japan) with the child seated and with the left arm resting on a table. In Norway, for children aged 7–12 years, the mean systolic blood pressure has been found to be between 95-115 mmHg, and we use this as a reference for our data[6]
Results section (lines 183–201)
- It is said that height and body mass increased normally with age, as 6th graders were 29 cm taller than 1st graders and weighed 18.7kg more. Simply suggesting descriptive statistics, but I recommend other statistical method, such as sampling significant variables and comparing it using t-test to find something significant result.
We acknowledge this comment on normal development, and have adden a reference to this.
This development does not deviate much from other developmental data of Norwegian children. [23].
Regarding the statistical comparison, we have added a p-value for the HDL results, and also added t-test analysis to the Statistics section. The aim of the study is not to compare groups, but to describe some important health variables in a young Norwegian population. We have thus avoided to emphesise t-test differences across sex or age.
Discussion section (lines 224–341)
- no in-depth discussion such as a comparison with advanced studies or mechanisms. It simply suggested the utility of radar graph, but there should be comparison of advanced studies, or other plot and graph. Also, the relationship between those plot or graph and measured variables should be described.
We agree that parts of the Discussion lacked depth. We have now included text and references through out the Discussion to counter this. Also, we have included information on the feasibility of Radar plots in Health Sciences, but the aim of our study is not to compare, but to describe.
Minor concerns
(1) lines 231 and 245, (2) lines 263 and 275, (3) lines 293–305, (4) lines 307–315, and (5) line 317-329
- Comparison with advanced studies should be added in various aspects, especially with regard to mechanism and adult health status.
We have tried to improve the lines mentioned. However adult health is not an aspect of this study

Reviewer 2 Report
This manuscript shows very useful and practical way to discover a present health status of certain population group. It can be used for both, individual cases and groups.
In this study the population was recruited from the Health Oriented Pedagogical Project (HOPP), a controlled longitudinal school-based physical activity intervention programme registered as a clinical trial (ClinicalTrials.gov Identifier: NCT02495714).
The results showed that the boys showed better endurance and 15 recorded more moderate to vigorous physical activity than the girls. The activity level dropped 16 from age 6 to age 12 for both sexes. The girls showed a lower systolic blood pressure compared 17 with boys. Self- rated quality of life was high among boys and girls.
However, I still have one suggestion to be made before further consideration:
In methods part of the manuscript, QOL (Quality of life method) should be better described: reliability, validity of the test, number of items, etc.
Author Response
We thank You for a kind review and agree with Your comment on the ILC description. and have added the following paragraph:
In any health profile, a variable describing an individual’s well-being should be included. Self-reported quality of life (QoL) was assessed with the Norwegian version of the Inventory of Life Quality in Children and Adolescents, (ILC)[16]. This is a seven-item inventory that uses 5-point Likert scales to score the answers. Children below 11 yr are advised to do the test by interview. We did this by having the children filling out the questionnaire themselves, under supervision. The inventory was given both to children, and to parents. The parental score is not presented here. Highest achievable score, 100, was used by us as the reference score. Kristensen and Hove [17] has been evaluating the internal consistency of the Norwegian version of ILC and found it to be acceptable (Cronbach α 0.63. to 0.76). Reliability was found to be good, 0.72).

Reviewer 3 Report
The topic of the manuscript “Presenting Health Status in Children Using a Radar Plot” seems an interesting issue and may be interest to readers of Sport. However, the main problem of the manuscript is the difficulty to evaluate the utility of this tool to present this kind of data. For this reason, the authors should be avoided some message about the utility of the tool because they did not evaluate it. Although the paper is very interesting, there are some minors concerns:
- In the introduction, it establishes the context and significance of the research being conducted by summarizing current understanding and background information about the topic, stating the purpose of the work in the form of the research problem supported by a hypothesis or a set of questions, briefly explaining the methodological approach used to examine the research problem, highlighting the potential outcomes your study can reveal, and outlining the remaining structure of the paper. The authors should be revised this information to add some extra information about the potential use of the radar plot in health science and introduce information about this manuscript add to the current evidence in this area.
- In the methodology section, the authors should clarify if the data used to build the radar plot were collected at baseline or follow-up.
- The authors should include the limitations and strengths of the study in the discussion section.
- A conclusion of the study should be included at the end of the discussion.
Author Response
We thank You for the pertinent comments:
- The authors should be revised this information to add some extra information about the potential use of the radar plot in health science and introduce information about this manuscript add to the current evidence in this area.
We agree on this comment, and have included a description of the plot and it's use.
The plot is not uncommon in health research, Minnesota Department of Health discusses the Radar plot and suggest it can be used for 1) displaying important categories of performance, and define full performance for each category, 2) show gaps between current and full performance, 3) capture a range of perceptions about performance, 4) provide data to support priorities for improving performance[4]. Our study looks at the feasibility of using the radar plot to describe a child population on various variables connected with cardiovascular disease (CVD) health, as this is not common to do.
2. In the methodology section, the authors should clarify if the data used to build the radar plot were collected at baseline or follow-up.
We agree and have added a description of the data collection:
The variables were collected according to Table 1 in spring 2015, which is the baseline results.
3.The authors should include the limitations and strengths of the study in the discussion section.
We acknowledge this comment and have added this to the end of the Discussion:
Strengths and limitations
The strength of this study is the large number of participants and the high number of carefully chosen variables collected. The weaknesses are the large number of persons that took part in the data collection and the use of some self-report data from children.
4. A conclusion of the study should be included at the end of the discussion.
We agree and have made this conclusion:
Conclusion
The investigated population was generally in good CVD health. A Radar plot was feasible to display a large number of health-related variables in a child population. The plot was useful in displaying both group data and cases. The presentation with a radar plot can help to identify children, or groups, in need of early intervention and may be used to investigate health tracking through childhood to adolescents and adulthood. If such a presentation is found useful, an expert panel should decide which variables and reference values to be used, any weighting of variables and how the data should be presented.

Round 2
Reviewer 1 Report
I accept your manuscript.
Best wishes!
Author Response
Thanks!
